

# Measurement of isoprene nitrates by GCMS

Graham.P. Mills[1], Glyn.D. Hiatt-Gipson[2], Sean.P. Bew[2] and Claire. E. Reeves[1]

[1] Centre for Ocean and Atmospheric Sciences, School of Environmental Sciences, University of East Anglia, Norwich, NR4 7TJ, UK

[2] School of Chemistry, University of East Anglia, Norwich, NR4 7TJ, UK.

*Correspondence to*: Graham Mills (g.mills@uea.ac.uk)

**Abstract.** An instrument based on gas chromatography/mass spectrometry (GCMS) and the associated calibration methods are described for the speciated measurements of individual isoprene nitrate isomers. Five of the primary isoprene nitrates formed by reaction of isoprene with the hydroxyl radical (OH) in the Master Chemical Mechanism are identified using

known isomers on two column phases, and are fully separated on the Rtx-200 column. Three primary isoprene nitrates from the reaction of isoprene with the nitrate radical ($NO_3$) are identified after synthesis from the already identified analogous hydroxy nitrate.

## 1 Introduction

On a global scale, isoprene is the most important biogenic volatile organic compound (VOC) in the atmosphere, with its

emissions accounting for 1/3 of the global total VOC emissions (i.e. natural and anthropogenic combined) (Guenther et al. 2006). It is emitted primarily by vegetation and mostly during the daytime. Being an unsaturated compound, it is readily oxidised by the OH and $NO_3$ radicals and ozone ($O_3$). The reaction with OH dominates during the daytime to produce eight isomeric hydroxyl peroxy radicals, which like other organic peroxy radicals ($RO_2$) radicals react with nitric oxide (NO). For isoprene-derived peroxy radicals, in common with most $RO_2$ radicals, the minor branch of the $RO_2$ + NO reaction yields

stable organic nitrates ($RONO_2$) as a product. At night when $NO_3$ concentrations are higher, $NO_3$ oxidation is believed to become the dominant loss process for isoprene (Perring et al. 2009a) and because of the higher estimated yields, reactions with $NO_3$ are expected to contribute more than 50 % of the total isoprene nitrates (INs) formed (Horowitz et al. 2007; von Kuhlmann et al. 2004).

The responses of different global chemistry transport models, in particular the production of ozone in the models, are sensi-

tive to the isoprene reaction schemes and yields used (e.g. (Fiore et al. 2005; Wu et al. 2007)) and recent studies suggest that the treatment of isoprene nitrates (INs) in the different models is the largest cause of these discrepancies (e.g. Emmerson and Evans, 2009).

In addition to modelling studies, there have been a number of laboratory studies which have used a variety of analytical methods to estimate the total yield of isoprene nitrates from the oxidation of isoprene and reported total yields range between

4.4 % to 15 % (e.g. Chen et al. 1998; Lockwood et al. 2010; Paulot et al. 2009; Sprengnether et al. 2002; Xiong et al. 2015,





Schwantes et al. 2015) and recently there have also been some recent reported measurements of speciated isoprene nitrates in chamber experiments (Nguyen et al. 2014; Schwantes et al. 2015).

There have been some field studies of the OH-oxidation derived nitrates (Schwantes et al. 2015; Perring et al. 2009b; Giaco-pelli et al. 2005; Grossenbacher et al. 2001 and 2004; Werner et al. 1999), however these measurements and, until very re-

cently, the laboratory experiments described above have all been ultimately limited in their use for model improvement and evaluation by the difficulty in unambiguously identifying and quantifying individual isomers of INs, particularly in the field. Some progress in the identification, quantification and measurement of reaction rates of individual isomers has recently been reported (Lockwood et al. 2010; Lee et al. 2014; Jacobs et al. 2015; Xiong et al. 2016), however, until recently, the limited availability of pure isomers for calibrations, identifications and kinetic studies as well as suitable methods for the accurate

determination of the isomeric distributions in complex mixtures have greatly restricted progress.

Here we present the results of the CASMIN project (Comprehensive Analytical System for Measuring Isoprene-derived Nitrates) to synthesise pure isomers of the primary INs, both OH and $NO_3$ derived, and to develop a GCMS-based method for the calibrated, speciated measurement of IN isomers using negative ion mass spectrometry which is highly sensitive to organic nitrates (Worton et al. 2008; Tanimoto et al. 2000).

**2 Synthesis**

The compounds we have investigated in this study are shown in Fig. 1. For the hydroxy nitrates, we follow the same naming convention as Lockwood et al. (2010). The aldehydic nitrates are labelled similarly to the equivalent hydroxy nitrate where the oxygen atom and nitrate are in the same position in the molecule, except they have "–al" as a suffix.

Until the very recent report of the synthesis of an isoprene derived carbonyl nitrate (Xiong et al. 2016), the reported synthe-

ses of IN isomers were limited to three INs made in a non-specific process that produced complex mixtures of the three INs and other products. The number of possible reaction mechanisms responsible for the IN formation and the mixture produced meant that post-separation identification of the INs was required. In contrast, our general synthetic approach was to build the isoprene nitrates in stages, which allowed us to design and assemble known carbon skeletons and use protecting group strategies as well as late-stage nitration under mild conditions to ensure selectivity in the location and extent of nitration and thus

yield individual, unambiguous isoprene nitrate isomers. We report the synthesis and purification of acetone nitrate (NOA), Z-(1,4)-IN, E-(1,4)-IN, Z-(4,1)-IN, E-(4,1)-IN and (4,3)-IN in detail in Bew et al. 2016 (the NMR data are also included in the supplementary information of this current paper) while the attempted syntheses of the remaining molecules in Fig.1 are described here.

As reported in Bew et al. the attempted synthesis of (2,1)-IN failed to produce a purified nitrate despite the synthesis of a

promising precursor. This was due to the surprisingly low reactivity of the precursor to nitration, despite a number of methods being applied, and the difficulties in separating the reaction products after nitration. Despite these problems, the initial post-nitration mixture was subjected to further steps in an attempt to produce (2,1)-IN. The synthetic methods, analytical





data and identification reasoning are included in the supplementary information. Because separation of the products was not achieved, interpretable NMR spectra were impossible to obtain but the headspace of this mixture, analysed using a chemilu-minescence (CL) system (see Sect. 5), showed the presence of $NO_Y$, whilst analysis with the GCMS showed a single volatile component that displays fragment ions with m/z 46 and m/z 62 with NI and a large m/z 46 with EI, suggesting that the ob-

served component is nitrated. Since identification is ambiguous, this compound will be referred to as species X.

The carbonyl nitrates in Fig. 1 were produced by oxidation of the corresponding hydroxy isoprene nitrate with manganese dioxide in acetonitrile, with purification by flash chromatography on silica gel. Based on GCMS headspace analysis, E-(1,4)-al-IN was produced as a single isomer from E-(1,4)-IN whilst the oxidation of Z-(1,4)-IN (probably as a mixture with E-(1,4)-IN) produced Z-(1,4)-al-IN as a mixture with E-(1,4)-al-IN. A synthesis of Z-(4,1)-al-IN was performed for identifica-

tion, but no further attempts to purify it from its parent alcohol were undertaken.

We were unable to obtain NMR data of purified Z-(1,4)-al-IN and Z-(4,1)-al-IN isomers as they appear to decompose or pol-ymerise rapidly, either during purification or in the NMR tube. The resulting spectra were broad and uninterpretable. We did however manage to obtain the [1]H NMR of the purified E-(1,4)-al-IN isomer (included in the supplementary information) and it is in excellent agreement with that of the same isomer very recently synthesised by Xiong et al. (2016).

**3 Identification, retention times and separations.**

**3.1 Retention times**

The retention times and mass spectra of the INs were determined by injecting approximately 20 µl of diluted headspace va-pour directly onto the column using a stainless steel 6-port Valco valve and a sample loop made from Sulfinert capillary tub-ing. Our initial characterisation attempts used a 60 m Rtx-1701 column. However, we had great difficulty in observing the E-

(1,4)-IN and Z-(1,4)-IN isomers. The chromatograms of these two isomers, when peaks were visible at all, showed signs of on-column decomposition such as baseline distortions and very asymmetric peak shapes, neither of which were observed with other isomers. Lowering the column temperature improved the peak shapes and reproducibility somewhat. The faster elution of these isomers on the 60m Rtx-200 column also improved results. The use of shorter columns reduced this problem yet further, with the isomers eluting after spending less time on-column and at a lower average temperature than on the long-

er columns. All identification, separation and calibrations reported here were thus performed with 30m long, 0.32 mm ID silica columns with 1 µm stationary phase thickness.

The (4,3)-IN, E-(1,4)-IN, E-(4,1)-IN, Z-(4,1)-IN , species X, E-(1,4)-al-IN isomers and acetone nitrate (NOA) retention times and mass spectra were recorded using individual isomers. The others were determined from mixtures of two isomers, one of which was already identified from a pure isomer. The retention time measurements were performed on two columns

of the same length, diameter and film thickness, but with different column phases. Both columns were operated under the same constant flow of He carrier of 4.0 ml min$^{-1}$ and the same oven program of 35 $^\circ$C (hold 3 min), increasing by 15 $^\circ$C min$^-$



$^{1}$ to 70 $^{\circ}$C (hold 1 min), +3 $^{\circ}$C min$^{-1}$ to 110 $^{\circ}$C (hold 15 min), +5 $^{\circ}$C min$^{-1}$ to 180 $^{\circ}$C (hold 7 min). The retention times on both columns are shown in Table 1.

### 3.2 Mass spectra

Electron-capture NI mass spectra (240 eV and argon buffer gas) were recorded in scan mode up to m/z 250 , EI mass spectra

(70 eV) were obtained up to m/z 250. The exception to this was compound X where the EI and NI mass spectra were measured up to m/z 400. These mass limits were chosen to be above the maximum expected mass of likely synthesis impurities, such as partially reacted precursors and dinitrates, whilst still retaining good sensitivity and scan rates.  Figure 2 shows the EI and NI mass spectra for four INs which represent the range of mass spectra obtained in this study. Full EI and NI mass spectra for all the compounds synthesised in this study are in the attached supplementary information. No ions above m/z

150 were observed for any of the synthesised compounds in this study.

### 3.2.1 Hydroxy nitrates

The EI mass spectra of the hydroxy nitrates are all similar and show fragmentation patterns that resemble pentenols, though with no sign of the molecular ion. The m/z 71 ion is most likely associated with the loss of the vinyl $CH_2ONO_2$ group – a common bond cleavage amongst the structurally analogous pentenols. Further fragmentation of the m/z 71 ion by loss of

$CH_3$, H, OH and $H_2O$ would result in the ions of masses 56-53. The masses 84, 83 and 82 are likely the result of $NO_2$ loss followed by further fragmentation of the mass 101 fragment by loss of OH, $H_2O$ and of both OH and $H_2$ respectively or by loss of $NO_3$ and subsequent loss of H atoms. The m/z 46 ion is common to all the IN investigated in this study, presumably the $[NO_2]^{+}$ ion, though species X has a much higher abundance than the other compounds.  Attempts to observe the molecular ion using Positive Chemical Ionisation (PCI) with methane as reagent gas resulted in poor sensitivity and mass spectra

that were very similar to those from 70 eV EI. Using 10 eV EI gave fewer fragments and yielded slightly higher abundances of larger fragments but as for PCI, showed no detectable molecular ion. Similarly NI mass spectra of the different IN isomers generally show the same ion fragments, with m/z 46 ($NO_2^{-}$) usually being the most abundant, behaviour which is comparable to simple alkyl nitrates (Worton et al. 2008). The m/z 99 and 101 ions are also prominent and are probably the organic fragments formed following $NO_2$ loss. The high abundance of these ions is entirely consistent with alkyl nitrate NI mass spectra

in which the organic fragments from larger nitrates tend to have higher abundances. For simple alkyl nitrates, ions of m/z corresponding to $[RO]^{-}$ fragments are rarely observed although fragments  with m/z corresponding to $[RO-H_2]^{-}$ are commonly observed (Worton et al. 2008) - usually presumed as elimination of $H_2$ from the α / β hydrogens. In contrast, both $[RO]^{-}$ and $[RO-H_2]^{-}$ ions are observed in INs, even in (Z)- and (E)- isomers where $H_2$ elimination pathways are not obvious. The proportions of m/z 99 and m/z 101 for E-(1,4)-IN are quite different to Z-(1,4)-IN, Z-(4,1)-IN, and E-(4,1)-IN, despite hav-

ing no more obvious source of $H_2$ loss than the others. Likewise (4,3)-IN produces mainly m/z 101, despite having suitable hydrogens available for elimination.





The EI mass spectrum of species X is very similar to the known INs, although it shows m/z 76 as a minor ion, presumably $CH_2ONO_2^+$, whilst the other synthesised isomers do not show this ion at all, only a m/z 71 fragment that would result from the same bond homolysis. As noted above, the relative abundance of m/z 46 is much higher in species X than the other nitrates. The NI mass spectrum of species X is quite different to the other INs in that it shows only two significant ions, presumably $NO_2^-$ (m/z 46) and $NO_3^-$ (m/z 62), with m/z 99 and 98 making up less than 1 % of the ions formed and m/z 101 not detectable. It is worth noting that Schymanski, et al. (2009) report that prediction of detailed mass spectra based on structure alone is not reliable.

### 3.2.2 Carbonyl nitrates

The EI mass spectrum of NOA shows m/z 43 as by far the most abundant ion resulting from the typical carbonyl α-cleavage to yield the $CH_3CO^+$ ion. m/z 76 is likely $CH_2ONO_2^+$, the other possible ion from the same bond rupture. Unlike the mass spectrum of acetone itself, no molecular ion (M) was observed.

The aldehydes typically show masses for $[M-NO_2]^+$ (m/z 99) and $[M-NO_3]^+$ (m/z 83). These ions may also lose CO to yield m/z 71 and 55 which are both common ions. Losses of 1 or 2 hydrogen atoms from these ions would yield m/z 98, 97, 82, 81, 69 and 53, the other commonly observed fragments. As for the hydroxy nitrates, PCI and 10 eV EI mass spectra showed no evidence of a molecular ion.

The NI mass spectra of NOA, as is typical of low molecular mass nitrates, yields m/z 46 as the major ion with only low abundances of the organic fragment from the same bond rupture (m/z 73). The m/z 71 fragment is presumably the result of further H atom or molecular $H_2$ loss from the m/z 73 fragment, although the nature of this loss process and the resulting ion are unclear. Unlike the small alkyl nitrates, m/z 62 is seen, albeit at low abundance.

The most distinctive characteristic of the NI mass spectra of the aldehydes are the high abundances of m/z 62 and a high proportion of m/z 98. The m/z 98 ion is most likely to be either the result of further H atom loss from the m/z 99 ion resulting from $NO_2$ loss, or by direct elimination of $HNO_2$ from the molecular ion, which itself is not seen.

### 3.3 Separation and photochemistry experiments

To test the potential for separation of mixtures of isomers in complex matrices with each column, a number of qualitative photolysis experiments were performed using 10L Tedlar sample bags. Several mixtures containing isoprene (approx.1 µl), NO (approx. 50 ppb), humid synthetic air (50 % RH) and a radical source (1 µl of ethyl nitrite in ethanol or t-butyl nitrite) were prepared in the Tedlar bags and exposed to ambient sunlight (clear, winter midday conditions for southern UK) for approximately 5 minutes. The contents of the bags were sampled (50 ml samples trapped using the heated capillary inlet and sampling methodology described in sect. 4) and analysed by NI-GCMS in SIM (selected ion monitoring) mode with only a small number of ions monitored. The different nitrites used as radical sources had no observable impact on the experiments. Similar mixtures containing no isoprene were also prepared and exposed to ambient sunlight as controls for comparison. No





particular effort was made to control or quantify the concentrations of components in the bags, or to measure the solar irradiance.

The chromatograms obtained without isoprene present contained only a few components and, with the exception of a small number of peaks on m/z 46, the retention window of the INs showed no significant peaks containing any of the m/z 62, 98,

99 and 101 ions. The chromatograms obtained when isoprene was present during photolysis are shown in Fig. 3 and contained many additional peaks throughout the chromatogram. Most of these new peaks had m/z 46 as their sole ion (of the ions we monitored) suggesting that they were products that contained the $-NO_2$ moiety and thus are likely to be some kind of nitrated organic compound. Figure 3a shows the total ion chromatogram (TIC) calculated from the sum of all monitored ions. The complex chromatogram means it is impossible to reliably identify any particular isoprene nitrate using the TIC (or

even only the m/z 46 ion which in this case constitutes the majority of the TIC signal) under these conditions. The use of IN fragment ions other than m/z 46 gave significantly simpler chromatograms with far fewer peaks and Figs. 3b and 4 show composite chromatograms comprised of these ions for analyses of similar photochemistry experiments on the two column phases used in this study. It is clear from these figures that using selected ions, isoprene nitrates can be identified in a complex mixture with either column although it is worth noting that the use of the Rtx-200 column allows the separation of the

E-(4,1)-IN and the Z-(1,4)-IN isomers which co-elute on the Rtx-1701.

The ions used in Figs 3b and 4 for each compound were chosen to give the least ambiguous identification and quantification of each IN within the samples from the bag photochemistry experiments. To estimate the relative proportions of isomers present in the bag, the quantifying ion's fraction of the total mass spectrum for that isomer was used as a scaling factor to correct for sensitivity differences. The results from the different experiments and columns are slightly different, but in both cases we

observe that (4,3)-IN represents $\geq$ 50 % of the observed INs.  In addition to the hydroxy-nitrates, it is also evident that aldehydic nitrates are formed under the conditions employed and in the Master Chemical Mechanism (MCM) such nitrates are represented by a lumped species (NC4CHO), which is only formed from $NO_3$ addition to isoprene.

The chromatograms of the bag photochemistry experiments also clearly show a component that has the same retention time and major ions as the synthesised species X and which is formed only in the experiments when isoprene and NOx are present

suggesting that species X is indeed an isoprene-derived nitrate. The peak marked (*) in figures 3 and 4 is formed during the bag experiments and the retention time and ions are identical to a small impurity in our synthesised species X, which has a very similar EI mass spectrum (in supplementary information) and an identical NI mass spectrum to species X.

## 4 Sample trapping and conditioning

Because of the reactive nature and low volatility of INs, all sample lines, valves and transfer lines were heated to 120 $^{\circ}$C to

reduce adsorptive losses and memory effects. Additionally, where possible, sections of GC columns were used as transfer lines to keep as many surfaces as inert as possible. Because of the improved separation, and faster elution times afforded by the Rtx-200 column, all the sample trapping and conditioning tests were performed using the Rtx-200 column.





The sample trap used to collect the data in this study was a ¼ " glass thermal desorption (TD) tube packed with 3 cm of 60/80 Tenax TA adsorbent, held in the tube with a glass wool plug. During sample trapping the trap was held at 35 °C and desorption off the trap onto the column was achieved with a reversed gas flow at 150 °C for 4 min. All samples were trapped at a constant flow of 40ml min$^{-1}$ controlled by a mass flow controller. No additional sample conditioning was used other than

to flush the trap with dry nitrogen for 1 minute before injection to reduce the oxygen in the tube. The TD tube was held between two siltek-treated stainless steel unions with PFA ferrules and was connected to a 6-port, stainless steel Valco valve with 15 cm of 0.53 mm ID MXT-1701 column maintained at 120 °C.

Injection temperatures of 150 °C and for a period of more than 3 minutes were used as it was found that, below these limits, re-heating the sample trap a second time yielded observable quantities of IN, indicating that complete desorption had not

occurred on the first heating . Higher injection temperatures were not studied because, as mentioned earlier, Z-(1,4)-IN and E-(1,4)-IN show signs of significant decomposition on the 60 m Rtx-1701 column at 150 °C (albeit they are exposed to those temperatures for considerably longer during separation on the column than during an injection). This trapping method is similar to that used by Grossenbacher et al. (2001, 2004) in the use of non-cooled Tenax adsorbents, although the trapping and desorption temperatures reported here are an intrinsic part of our different methodology for dealing with oxidants and to ac-

commodate the thermal decomposition characteristics of INs.

To test the trapping method, materials and conditions for linearity and potential breakthrough of INs on the trap, six samples with volumes between 120 and 480 ml of a mixture of INs and ethylhexyl nitrate (C$_8$ alkyl nitrate) in a temperature controlled drum (see Sect. 5) were trapped and injected onto the column. Over the range covered by the test volumes, the observed MSD signal was linearly proportional to the trapped volume for the INs (figure included in supplementary infor-

mation) which indicated that the breakthrough volumes at 35 °C are greater than 480 ml. The linearity with trapping volume also indicates that the compounds do not decompose on the adsorbent to any significant degree over the trapping period. Confirmation that decomposition of the adsorbed INs was not significant over the trapping period was obtained when 160 ml samples trapped and held on the trap at 35 °C for a further 20 minutes were indistinguishable from those trapped and injected immediately.

The reproducibility of the trapping and injection method was assessed by analysing seven consecutive 200 ml samples of the same mixture from the temperature controlled drum (graph in supplementary information). Excluding species X, the standard deviation of the seven measurements range between 3 % for the C$_8$ alkyl nitrate and 6.6 % for E-(4,1)-IN with a mean of 4.9 % for the five isoprene nitrates. Species X has a much larger standard deviation than the other components (10.3 %), despite it behaving in a similar manner to the other IN during the linearity tests. The reactive nature of INs means that condi-

tioning of surfaces in the system is a potential issue that may impact on the precision and uncertainty in real measurements at low mole fractions. Repeated measurements of the INs at low absolute abundances would show evidence of any such conditioning occurring within the system. To test this, six 50 ml samples of the IN mixture in the temperature controlled drum were trapped and injected onto the GCMS system after it had been left unused for two weeks. We did not observe any in-



crease in the GCMS response with the increasing number of sample injections. Nor did we see different responses for 50 ml samples analysed before and after the sampling of much larger volumes.

## 4.1 Oxidant impacts

Isoprene nitrates contain both an unsaturated C=C bond and a hydroxyl group, which provide sites for attack by oxidants such as $NO_2$ and $O_3$. To investigate the potential impacts of sampling INs in an oxidant-rich environment, 200 ml samples from a mixture of IN (and a $C_8$ alkyl nitrate) were collected normally and then, before injection, an equal volume of one of $NO_2$, $O_3$ (at 100 ppb (ppb; nmol/mol)) or clean synthetic air was sampled on to the same trap. The trap contents were then flushed for 1 min with nitrogen to remove any air or oxidant from the sample prior to injection and the sample injected normally. Two consecutive samples of the IN mixture plus air were compared to two consecutive samples of the IN mixture plus oxidant for each oxidant. At a trap temperature of 35 °C, there was no discernible difference between the samples. In contrast, at a trapping temperature of -15 °C, there were reductions of approximately 25 % in the INs (and no corresponding change in the $C_8$ alkyl nitrate in the mixture) when $NO_2$ or $O_3$ was trapped compared to the synthetic air controls. This indicates that at 35 °C the oxidants are not co-trapped with the INs and have little impact on their analysis and that prior conditioning of the sample is unnecessary.

## 4.2 Impact of humidity

Lee et al. (2014) report that when trapping three IN isomers directly onto the analytical column at -20 °C they observed evidence of heterogeneous reactions as a result of co-trapping of water, something which will be more important for field measurements than chamber studies in dry air. In this work, we trap at above-ambient temperatures on hydrophobic adsorbents which will prevent the concentration of water, and thus hydrolysis of the IN on the trap. On the very few occasions where water (m/z 18) was monitored on the EI MSD it remained at background levels throughout the chromatogram, suggesting that we do not collect water on the trap, although it remains a slight possibility that any water possibly concentrated on the trap has eluted very quickly and cleanly in the time allowed for residual air from the trap to leave the column before the MSD is turned on.

## 4.3 Other trapping methods

Cryogenic trapping at -150 °C using empty silcosteel tube, and at -50 °C using sections of MXT column and Sulfinert traps with Rtx-1 coated column packing material were also tested as potential trapping materials and methods in this study. Cryogenic trapping gave comparable precision to that obtained with Tenax traps when analysing the same nitrate mixture multiple times, but instrument sensitivities determined with different IN dilutions using cryogenic trapping gave poor reproducibility, probably due to the different $NO_2$ concentrations in each IN dilution. In addition, water would also be effectively trapped meaning that scrubbers for $NO_2$, $O_3$ and water would be required for reliable quantitative analysis of INs. The reactive nature and low volatility of the INs would mean finding suitable scrubbers would be very difficult or impossible. For example man-





ganese dioxide ($MnO_2$), a common $O_3$ scrubber, is utilised in our syntheses of the aldehydic nitrates to oxidise the corresponding OH-containing IN to the aldehyde, and so is highly unlikely to be a suitable $O_3$ scrubber. The use of Rtx-1 coated column packing material (10 cm 1/8" Sulfinert loop packed with 20 % Rtx-1 on 100/120 Silcoport W, Restek) was also in-
vestigated. At 30 $^\circ$C, there was little impact from $NO_2$ and $O_3$ on the sampled IN, however the breakthrough volumes of the more volatile INs (such as (4,3)-IN) were found to be 100-200 ml. Trapping at -15 $^\circ$C improved the breakthrough volume to >500 ml, but $NO_2$ and $O_3$ were then found to adversely affect the analysis.

**5 Calibration methodology**

Our initial attempts to calibrate INs utilised a vacuum line equipped with a calibrated volume to inject pure isoprene nitrate vapour at measured temperature and pressure either onto the GC column directly or into a known dilution volume (glass,
polyethene or aluminium), a method that has worked well for stable compounds. However, this proved to be impossible due to the reactive nature of the isoprene nitrates. The measured vapour pressures of the pure isomers was typically below 0.05 mbar at 25 $^\circ$C which would have allowed a single step dilution to ppt ($10^{-12}$ mol/mol) mole fractions, however the observed pressure in the system continually increased in the presence of the nitrates. Furthermore, the results of either direct injection to the column or sampling from the dilution drum gave very variable results. It is highly likely that adsorption and decompo-
sition in the vacuum line prevented the production of accurate and repeatable IN concentrations with this method.
We thus used the dilution of single isomers with synthetic air in drums to produce indeterminate mole fractions which were then accurately measured by thermal decomposition to $NO_2$ and a chemiluminescense (CL) measurement of the $NO_2$ as the basis of our calibrations.

**5.1 Chemiluminesence detector and measurements.**

Thermal decomposition at approximately 400 $^\circ$C in a quartz tube has been used to convert organic nitrates into $NO_2$ for detection by a number of methods (Day et al. 2002; Lockwood et al. 2010; O'Brien et al. 1998; Paul et al. 2009) and our system does not differ fundamentally from these. The quartz loop used in this study was 2 mm ID and the heated length was approximately 10 cm. At our sample flow rate of 30 ml min$^{-1}$ the residence time in the heated zone was in the order of 500 ms, sufficient time to give quantitative decomposition of organic nitrates to $NO_2$ (Day et al. 2002). The resulting $NO_2$ was deter-
mined by a simple and well-established luminol-based CL method (Maeda et al. 1980; Kelley et al. 1990). The $NO_Y$ content of the drum was determined from the difference in CL signals obtained when the sample passed through the same quartz tube when it was heated and when it was at room temperature, and assuming that all the organic nitrates thermally decompose to yield $NO_2$ quantitatively. The CL detector was itself calibrated against a $NO_2$ gas standard ($9.5 \times 10^{-6}$ mol/mol in nitrogen, BOC Spectra-Seal) diluted with synthetic air (BTCA 178, BOC) to mole fractions between 5 and 100 ppb. The measurement
precision of the CL detector based on signal variability and sensitivity during repeated calibrations was 5 % with a minimum overall uncertainty of 98 ppt, requiring that calibrations were performed on low ppb mixing ratios of nitrates rather than the



low ppt levels we expect in the real atmosphere. The overall uncertainty of the mole fraction of $NO_Y$ determined by the CL method, including the certified accuracy of the $NO_2$ standard is estimated at ± 11.7 % and the average overall uncertainty for the measurement of INs (including the GCMS precision and calibration uncertainties) is ± 14 %.

### 5.2 IN calibration results

For the calibrations a glass cube with 30 cm sides was constructed from laminated glass, with the contents mixed with a small brushless fan. The size of the cube allowed rapid internal mixing and allowed us to heat the cube in an oven to aid cleaning. During experiments, the cube's contents were shielded from light with aluminium foil on all sides.

Sub-µl quantities of an isoprene nitrate were introduced into the cube by wetting the tip of a fine stainless steel wire with the isomer and placing the tip inside the volume for 1-10 minutes and allowing it to evaporate. It was left for at least 2 hours to

equilibrate before it was analysed by CL. Further flushing with BTCA air and re-equilibration were performed until the observed $NO_Y$ mole fraction was approximately the desired value. Experiments with ethylhexane nitrate showed that complete mixing occurred in less than 10 minutes within the cube. However, it was found that the isoprene nitrate levels dropped rapidly even after this 10 minute mixing period and continued to decrease slowly for several days, although after 2 hours the rate of change was low enough that it was effectively constant over the time required for concentration determinations and

GCMS sampling of the cube and therefore did not adversely impact the measurements. The rapid initial losses and the subsequent slow decrease in IN concentrations occurred in 100 L aluminium drums and 15 L polyethene drums as well as the glass cube.

After CL analysis, the cube's contents were analysed immediately by GCMS and then were reanalysed by CL for $NO_Y$ content to quantify the effects of sample removal and decomposition during GCMS sample collection. Typically less than 1 % of

the cube's volume was removed during CL and GCMS analysis, and over the 10 minute period between the two CL analyses no difference in $NO_Y$ was observed within the precision of the CL measurement. As for the tests in Sect. 4, all GCMS calibrations of the INs used the Rtx-200 column because of its improved performance compared to the Rtx-1701.

The cube's contents were sampled by the GCMS at 25 ml min$^{-1}$ for 2-4 minutes through a heated capillary column inlet (0.53 mm ID Rtx-200 at 120 $^\circ$C), 5 cm of which was inserted through the cube's inlet port to sample the contents directly.

Figure 5 shows the calibration results using this method at four different mole fractions of Z-(4,1)-IN, and it is evident that the GCMS response is linearly proportional to the $NO_Y$ mole fraction measured by CL, and also that the GCMS sensitivity determined at each of these very different mole fractions is the same within the measurement precision of the instrument. The GCMS sensitivity of three isoprene nitrates and n-butyl nitrate were determined from at least three sets of measurements, and the results, shown in Table 2, indicate that the NI MSD is typically more sensitive to the INs than a simple alkyl

nitrate. This is consistent with the findings of Lockwood et al. (2010) who showed that a GC electron capture detector (ECD) system was more sensitive to INs than n-butyl nitrate.





Typically the limit of detection for n-butyl nitrate on our MSD-based system is < 0.05 ppt for 500 ml samples, and the higher S/N observed for the INs suggests that a similar LoD is, in principle, possible, although measurements of real air at a location and during a season where isoprene is abundant have not yet been made to verify this.

### 5.3 IN mixtures - stability

During the development of the calibration and trapping methods, it was observed that dilution drums (glass, aluminium and polyethylene) left for several weeks still contained appreciable concentrations of the INs unless they had been cleaned at elevated temperatures. It was also noted that the observed concentration of INs from the same drum was very sensitive to the temperature of the drum (See supplementary information for figure and details) suggesting that some form of equilibrium between adsorbed and gas-phase INs had established. In light of this temperature dependence, an aluminium drum was sub-

sequently insulated and controlled at $31\,^{\circ}$C (warmer than room temperature and cooler than trapping temperature) and this temperature-controlled drum was used in the precision measurements and sampling tests (Sect. 4).

For those tests, a number of isoprene nitrates and a stable $C_8$ alkyl nitrate (ethylhexyl nitrate) were introduced into the drum and left for 3 weeks to equilibrate before being analysed by GCMS. Based on our GCMS sensitivity to Z-(4,1)-IN, a crude estimate of the mole fractions of the isomers in the drum after the initial equilibration period were 1 ppb for Z-(4,1)-IN and

less than 100 ppt for the minor components. The drum mixture was sampled repeatedly over 3 weeks during repeated analytical tests, then again 1 month and 4 months later. The results of measurements made under identical conditions to the initial measurement are shown in Fig. 6. Over the initial measurement period, the ratio of most of the INs with $C_8$ nitrate show a slight decrease, the obvious exception being E-(1,4)-al-IN, which shows a larger decrease in the ratio presumably as a result of continued decomposition of the aldehyde. Fig. 6 also indicates that the IN:$C_8$ nitrate ratio continues to decrease over time,

suggesting that the INs are still decomposing slowly after 4 months.

### 5.4 Possible field sampling and calibration methodology

To calibrate the system during short, laboratory-based studies we can use individual INs and drum dilutions or aged, temperature controlled, and slowly changing dilutions of mixtures of INs that have had the mole fractions of the components individually determined. For fieldwork, a less cumbersome and more practical solution would be desirable. One method would

be to generate and measure a mixture of INs in a drum dilution and to sample the drum with a number of Tenax adsorption tubes identical to the main sample trap. The tubes can then be stored at -25 $^{\circ}$C, a temperature at which the nitrates are stable, and used in place of the main trap as an in-field multi-component standard. An alternative and complementary approach is to determine the sensitivity of the MSD to each IN relative to that of a stable organic nitrate (as we have done against n-butyl nitrate in this work) and then use an organic nitrate standard to calibrate for the INs indirectly.

The fact we use a Tenax adsorption tube as our sample trap without any need for sample pre-treatment means that such tubes should be suitable for general sample collection, provided the collected samples are stored at -25 $^{\circ}$C to prevent decomposition.




### 6 Comparison with earlier work

Until the very recently reported synthesis of Z-(1,4)-al-IN (Xiong et al. 2016) and our reported synthesis of five of the hydroxy isoprene nitrates by unambiguous means in Bew et al. 2016, the only published syntheses of isoprene nitrates were those of three IN isomers separated from mixtures reported in detail by Lockwood et al. (2010) and Lee et al. (2014). Both

groups used the same starting material but differ in the particular, but similar, nitration conditions employed, and it is quite probable that they synthesised the same isomers although they identify them differently. This highlights the potential ambiguity that results from identifying components separated from complex synthetic mixtures. The NMR spectra obtained for our synthesised E-(1,4)-IN, Z-(1,4)-IN and (4,3)- IN are identical to those obtained by Lee et al.(2014). As reported in Bew et al. (2016), Z-(1,4)-IN isomerises rapidly to E-(1,4)-IN, resulting in a mixture of the two isomers, a situation also reported

by Lee et al. (2014) who found that samples of Z-(1,4)-IN contained approximately 15 % of the E-(1,4)-IN.  Similarly, Lockwood et al. report similar behaviour for two of their isomers which are clearly separated by HPLC yet are mixtures when analysed by GC.

Our attempted synthesis of (2,1)-IN produced species X, a compound that has many characteristics expected from an isoprene nitrate and was produced from a starting material that should yield very few products. However, if species X is (2,1)-

IN then it elutes much later than the structurally similar (4,3)-IN, which would be in total disagreement to the reports of Nguyen et al. (2015) which shows that (2,1)-IN elutes before (4,3)-IN on the Rtx-1701 column. The Nguyen study used a synthesised standard (although the synthesis and analytical data are as yet unpublished) to determine the retention order and also, importantly, used soft ionisation method that measures the molecular ion. It therefore seems likely that species X is not the (2,1)-IN isomer although it may well be an isoprene-derived nitrate of some sort.

The bag photochemistry experiments in this study, while poorly constrained, produced isomer distributions that were very different to those typically produced by models (e.g Paulot et al. 2009) which predict the (1,2)-IN to be the major isomer. We observed (4,3)–IN as by far the most abundant isomer and we do not see any peaks near (4,3)-IN in the chromatograms that are strong candidates for an isoprene nitrate. Both Nguyen et al. (2014) and Xiong et al. (2015) using CIMS detection report that (1,2)-IN (identified from a mix of isomers generated in chamber photolysis experiments) elutes before (4,3)-IN on

the Rtx-1701 column (though again there is no currently published synthesis or analytical data for this isomer). The (1,2)-IN isomer is reported to be more reactive than the other IN isomers (J.D. Crounse, Pers. Com) and the most likely explanation for our unexpected yields is that under the particular trapping and/or analytical conditions used here the (1,2)-IN isomer is lost before detection.

Our observed elution order on the Rtx-1701 column for the hydroxy isoprene nitrates synthesised in this study are entirely in

agreement with those reported by Schwantes et al. (2015) using the same column phase.  Our observed elution window between the (4,3) and the Z-(4,1)-IN isomers and the elution order of the three aldehydic isoprene nitrates we synthesised in this study agree well with those reported by Schwantes et al. That study predicted elution order from those of the analogue alcohols and peroxides as well as theoretically derived yields, whilst we identified ours based on direct production from





known synthesised hydroxy nitrate isomers. The positive identification of E-(1,4)-al-IN by NMR increases our confidence in the identities of the other two aldehydes since they were synthesised by the same method, although without NMR there will remain some uncertainty in their identity.

**7 Conclusions and further work**

In the CASMIN project, we have synthesised five isoprene hydroxy nitrates using controlled synthesis routes and developed a simple GCMS method for their analysis in complex air matrices, including those with realistic relative humidity, although further improvements are needed to overcome our problems observing the (1,2)-IN isomer. We have identified the nitrates on two different columns and have shown that the use of an Rtx-200 column allows the separation of two isomers (E-(4,1)-IN and Z-(1,4)-IN) which co-elute on the more widely used Rtx-1701 column. In addition we have synthesised and identified

three carbonyl isoprene nitrates , one of which is unambiguously identified by NMR and that confirms the results of recent work. Furthermore we have demonstrated that they can be separated and measured as individual isomers in photochemistry experiments.

The unidentified species X, produced during an attempt at synthesising (2,1)-IN, is also formed during the same photochemistry experiments in which the INs were observed. The limited evidence available suggests that it is a nitrated species and

since it is only observed in photochemistry experiments when isoprene is present, it may be an isoprene-derived nitrate of some sort.

**Acknowledgements.**

We would like to acknowledge funding from the National Environmental Research Council (NERC) under grant number NE/J008389/1.



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





| column | Isoprene nitrate | | | | | | | | | |
|---|---|---|---|---|---|---|---|---|---|---|
| | (4,3) | Z-(4,1) | Z-(1,4) | E-(4,1) | E-(1,4) | Z-(1,4)-al | E-(1,4)-al | Z-(4,1)-al | NOA | Species X |
| Rtx-1701 | 26.1 | 36.5 | 39.4 | 39.3 | 41.2 | 27.0 | 28.3 | 33.7 | 15.5 | 45.4 |
| Rtx-200 | 16.7 | 22.4 | 23.3 | 25.1 | 26.5 | 22.9 | 24.5 | 29.1 | 13.7 | 38.8 |

Table 1. Retention times (minutes) for isoprene nitrates on two different stationary phases. Both columns are 30 m, 0.32 mm ID, 1μm phase thickness. The flow and oven conditions are the same for both columns. A constant flow of He carrier of 4.0 ml min$^{-1}$ was used and the oven program was 35 $^{\circ}$C (hold 3 min), increasing by 15 $^{\circ}$C min$^{-1}$ to 70 $^{\circ}$C (hold 1 min), +3 $^{\circ}$C min$^{-1}$ to 110 $^{\circ}$C (hold 15 min), +5 $^{\circ}$C min$^{-1}$ to 180 $^{\circ}$C (hold 7 min).





| Isoprene Nitrate | Sensitivity:   (s/n)/ppt of ion relative to n-butyl nitrate m/z 46 | | | |
|---|---|---|---|---|
|  | m/z 46 | m/z 101 | m/z 99 | m/z 62 |
| (4,3)-IN | 1.53 | 5.53 | 0.4 | 0.175 |
| Z-(4,1)-IN | 1.92 | 1.47 | 1.62 | 0.875 |
| NOA | 2.97 | - | - | 0.625 |

Table 2. Relative sensitivities of different INs to n-butyl nitrate, using different quantification ions. Relative sensitivity is
5   obtained from signal/noise (s/n) for each ion (per unit mole fraction), divided by the s/n of n-butyl nitrate (m/z 46) per unit
mole fraction.





Figure 1. Isoprene-derived nitrates investigated in the CASMIN project. The labelling scheme for hydroxy nitrates is the same as that of Lockwood et al. (2010) and for the aldehydes as described in the text. NOA is acetone nitrate.





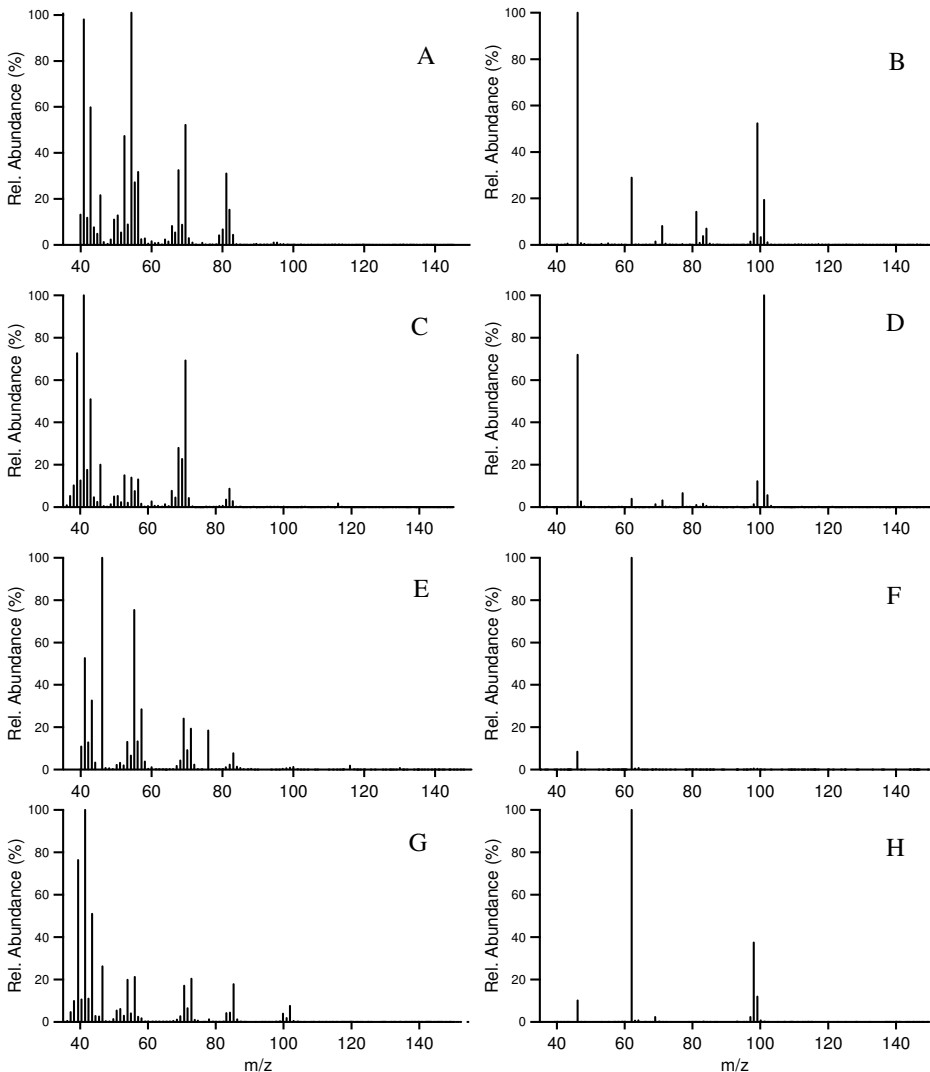

Figure 2. Mass spectra (EI left, NI right) of selected isoprene nitrates: E-(4,1)-IN (A and B), (4,3)-IN (C and D), species X (E and F) and E-(4,1)-al-IN (G and H). Mass axes are limited to m/z below 150 for clarity, however scans were to higher m/z (see main text) but no ions above this m/z were observed for any compounds. EI and NI mass spectra for all the synthesised

5 INs including more annotated versions of those shown here are in the supplementary information.





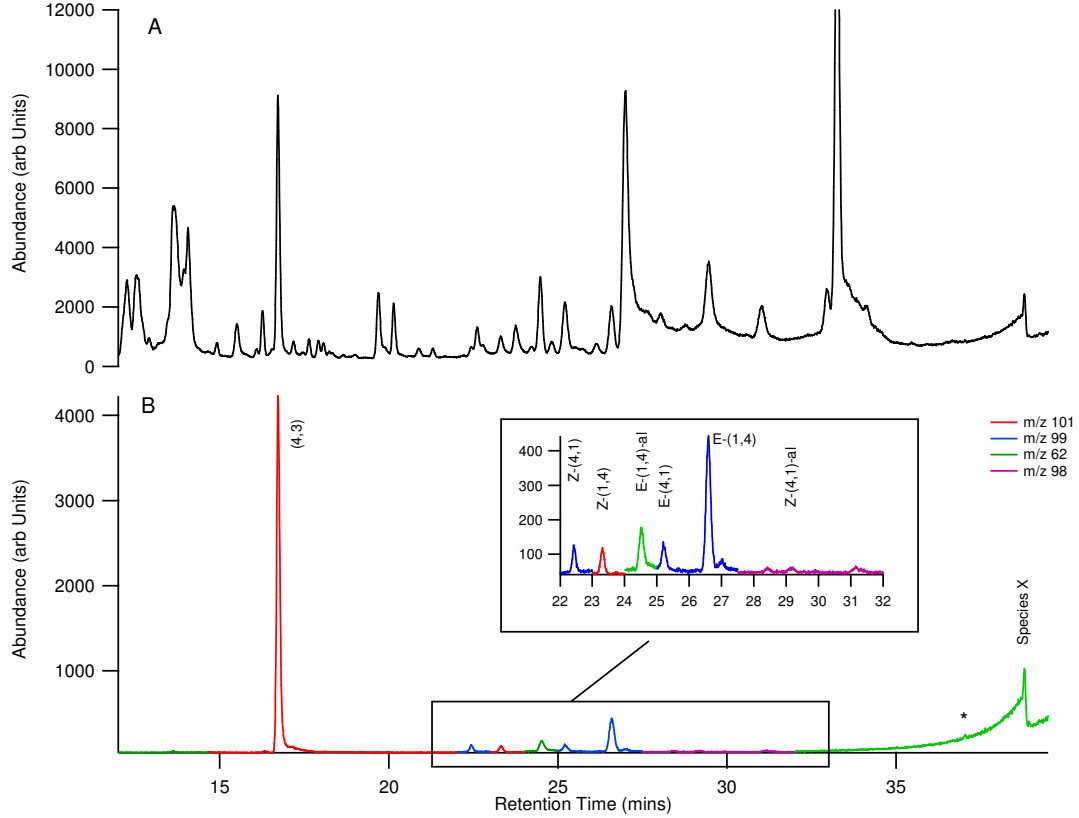

Figure 3. Results of a bag photochemistry experiment analysed on a 30 m, 0.32 mm ID Rtx-200 column using NI mass spectrometry. A) Total Ion Chromatogram. B) Composite extracted-ion chromatogram of the data in A). The small peak marked (*) has the same retention time and ions as the small impurity in the synthesised species X.





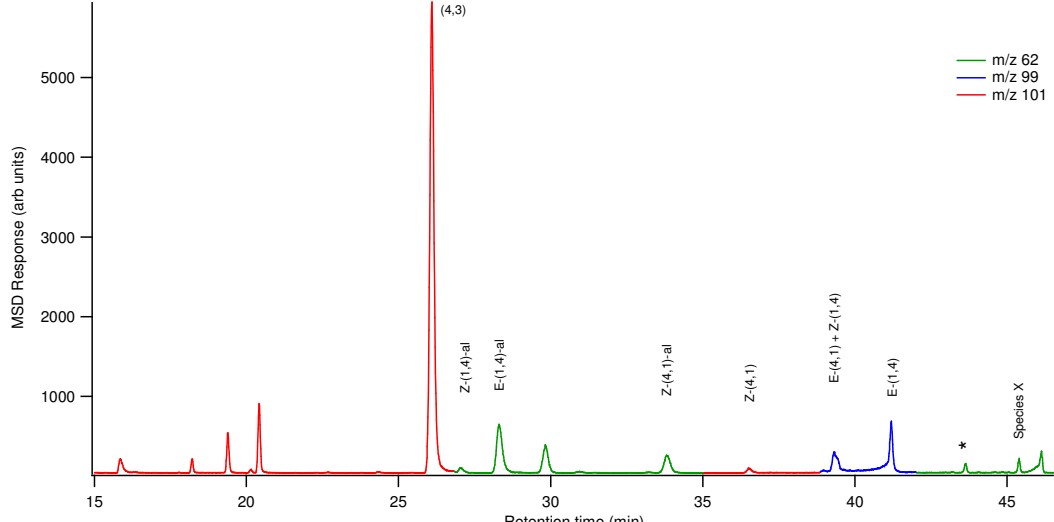

Figure 4. Composite extracted-ion chromatogram showing INs formed during a bag photochemistry experiment and ana-lysed on a 30 m, 0.32 mm ID, Rtx-1701 column. The conditions for the photolysis are similar to those used in Fig 4, and the analytical conditions are identical to those used for the Rtx-200 column used in Fig. 4. The E-(4,1) and Z-(1,4) isomers are not separated under the column conditions used. The peak marked (*) is the same retention time and major ions as the impu-rity in the synthesised species X




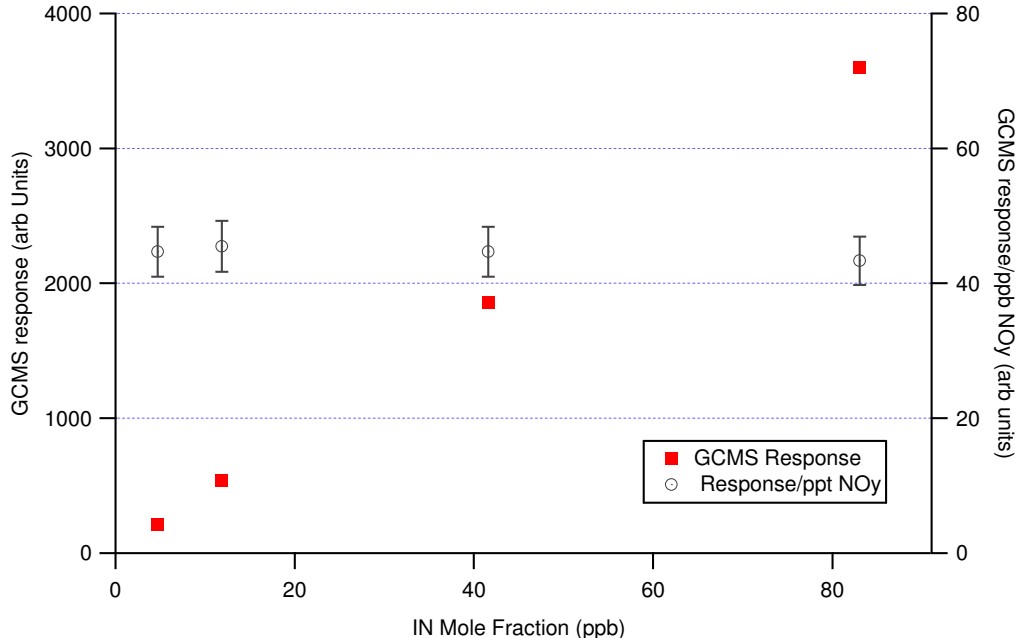

Figure 5. GCMS response and calculated GCMS sensitivity for Z-(4,1)-IN at different mole fractions (measured by CL).

The error bars shown are the combined CL and GCMS measurement precisions (± 8.3%) as discussed in Sect. 4.




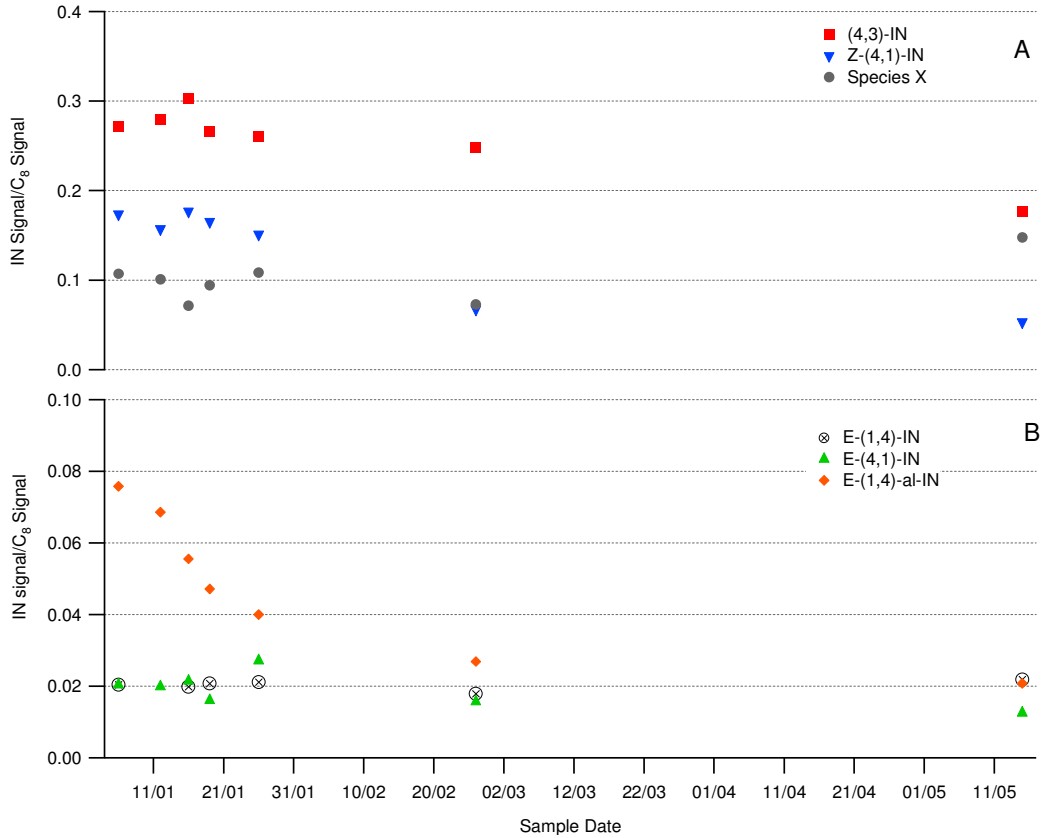

Figure 6. Ratios of INs to a stable $C_8$ alkyl nitrate in a temperature-controlled, fan-mixed 100L aluminium drum measured

5    over a period of 4 months. The drum contents were left for 3 weeks to equilibrate before the first sample.