# Peer review of "S1. Attempted synthesis of (2,1)-IN"

_Atmospheric Measurement Techniques, 2016_

## Referee Comment (RC1) · Anonymous Referee #3 · 26 May 2016

Summary: Isoprene nitrates (INs) are important components of ambient air particularly in isoprene-rich areas; however, they are difficult to measure and individually identify given the low and variable ambient concentrations of any individual isomer, their chemical reactivities, and high polarities. The authors describe how to synthesize various isoprene nitrates (IN) that are utilized in this study. The majority of the paper focuses on the development of GC-MS methods to fully resolve and positively identify each of 10 IN isomers using different chromatography columns, ionization sources, and calibration methods.

Recommendation: This paper is within the scope of AMT and is a valuable to the larger scientific community. I recommend publications with only minor revisions (see below).

Minor comments:

Abstract: This section should contain more information that what is presented here in

my opinion. It should provide the motivation for the study, a brief description of the method (or aim thereof), and a summary of the most relevant findings.

P1L9 : Add "in the presence of NOx" to describe how isoprene nitrates are formed.

Introduction: The words "recent" and "recently" are overused in the introduction. "Previous" is an alternative term that should be considered.

P1L25: There are an extra set of parentheses around the references

P2L26: Could you briefly elaborate on the "treatment of isoprene nitrates in the difference models"? Are they excluded, lumped, have different reactivities, etc.?

P2L25: It would help the reader if the list of species starting with acetone nitrate (NOA) were in the same order as that presented in Fig. 1 (i.e., (4,3)-IN, ..., ending in NOA).

P2L27: List the remaining isomers to be more direct and reader friendly.

P3L15: Consider changing this section header to "Identification of Isoprene Nitrates via GC-MS"

P3L16: Consider changing this section sub-header to "Chromatographic Methods"

P4L9: Figure S2.1 is really great. The authors should consider moving S2.1 into the main text in place of Fig. 2. In either figure (S2.1 and/or Fig. 2), it would be very helpful to put the molecular structures from Fig. 1 on each of the respective panels. It makes the mass spectra interpretation discussion (3.2.1 and 3.2.2) much easier to follow.

P5L23: Consider changing this section sub-header to "Sample matrix and photochemistry experiments"

P7L18: What is the volume of the drum used and the approximate concentrations of the analytes within the drum? How does the volume extracted for each sample (and sum of all samples) compare to the initial volume? Wall losses/effects would likely be more pronounced for lower pressure, lower concentration mixtures within a drum.

---

## Referee Comment (RC2) · J. D. Crounse (Referee) · 3 Jun 2016

This paper describes analytical methods for separating and measuring several organic nitrates which can be formed in during the oxidation of isoprene in Earth's atmosphere. Methods are put forward to allow measurement of these species in air, and demonstrated by analyzing products formed from photo-chemical reactions in a synthetic reactor with isoprene, air, NOx, and sunlight.

This paper is well-written and very appropriate for publication in the AMT journal after consideration of the comments listed below. This work taken together with the companion work published in Bew, et al, 2016, represents an important step forward for making progress in our understanding of isoprene chemistry in our atmosphere.

General comments:

1) The authors state that there were problems for the isoprene hydroxynitrate trans-

mission with longer columns (and warmer elution temperatures). Then it is stated the analyses were run on 30m instead of 60m columns, but it is not demonstrated that this completely eliminated the problem of hydroxynitrate decomposition or conversion. It is suggested here that the authors continue reducing their column length until results are reproducible. Additionally, I note that in our experience the 1-OH-2-ONO2 ISOPN isomer continues to suffer decomposition/isomerization problems in a metal-free low pressure GC-CIMS setup even on a 4-meter 0.53 megabore RTX-1701 column, with head-pressure at ∼150 mbar and tail pressure ∼40 mbar, under which it elutes at ∼60C. Additionally, and not without significant effort, we have not been able to trap the 1,2- externally and then transfer this quantitatively to the column without significant losses. Thus, our preferred method is to trap these compounds directly on the column phase at ∼ -15C. Care must be taken not to also condense water, as co-condensation of water and 1,2-ISOPN results in rapid hydrolysis of the 1,2-ISOPN.

2) Order of elution for the hydroxynitrate isomers. In addition to the results published in Lee, et al, 2014, we now have NMR-supported identification of the 1,2-, c-4,1-, and t-4,1- ISOPN isomers. We find the elution order on RTX-1701 column to be 1,2- , 4,3-, c4,1-, (t4,1 and c1,4 together), and finally t1,4. These chromatograms from isoprene oxidation experiments are shown in Nguyen, et al., 2014, and the elution order for the five primary isoprene hydroxyl nitrates identified here is in agreement. In addition, Xiong, et al 2015, using I- CIMS shows a chromatogram from an OH+isoprene experiment, with a similar chromatographic technique. One primary problem that has remained through the years with the analysis of isoprene hydroxyl nitrates appears to be the very high instability of the 1,2-ISOPN. It appears to be thermally instable, decomposing and isomerizing to t1,4 isomer even at temperatures as low as 60C. Given the structural similarity of the 2,1-ISOPN to the 1,2-ISOPN and the general elution relationship for 1,2 vs 2,1 hydroxynitrates for a series of alkenes (Teng, et al, 2015). I would suggest that the 2,1-ISOPN isomer should elute even before the 1,2-ISOPN. Based on this, I suggest that the late-eluting species reported in this work arises from something other than (or a conversion product of) the 2,1-ISOPN species.
3) The 1,2-ISOPN has a very fast hydrolysis rate. We've constrained (unpublished) its lifetime to be <5 seconds in D2O using NMR experiments. It is likely it also reacts rapidly in the condensed phase with other species, including itself. While, on a clean Teflon surface minimal H2O is adsorbed, it is possible that on a dirty Teflon surface, other low volatility hydrophilic material could be present, allowing for uptake of water, and thus allowing for a fast hydrolysis rate in a humid bag. I suggest that these experiments be tried with very dry air in a clean bag, using a GC method akin to those described in the references below (short column (2-4m), 8 sccm gas flow, sub-ambient pressure, such that 4,3-ISOPN elutes at <65C, no metal in sampling system or GC-system (PEEK/PTFE/PFA seem okay), trap analytes directly on column at reduced temperatures (-15C)), and see if the same results are achieved.

Specific comments:

Working from print version:

P4 L5: Can the authors describe the energetics of the chemical (NI and PI) ionization schemes?

P5 L24-26: Suggest writing the isoprene and nitrite amounts as a mixing ratio, as this will be more useful to the reader (∼20ppmv, each). P7 L19: Define MSD acronym.

P8 L29: Where did the NO2 come from? IN decomposition?

Section 5.1: It seem to me that important inherent assumptions in this method of calibration include: a) decomposition of IN to NO2 in the cold quartz tube (or entire CL system) is small; b) reaction of IN with luminal is small. It would be useful to discuss the validity of these assumptions in more detail. It would be useful to include a figure of a CL experiment in the SI so the reader can get a feel for the time constants inherent with this method. What is the typical 'cold' CL signal compared to the IN signal?

Section 5.2: Is it proper to interpret this procedure as contents of the cube were 'directly' sampled to the MS through a short piece of heated column? It would be natural

then to compare the sensitivities determined in this manner to sensitivities determined in a similar manner with the Tenax trapping using the same CL reference, as this is needed to show that the transmission of the trapping method is high, and suitable for field samples.

P11 L26, 31: It is stated here that the IN's discussed in this manuscript are stable on Tenax stored at -25C. This statement could use further support.

P12 L25-28: The 1,2-IN could also be lost through hydrolysis in the humid bag.

P13 L13-16: Is it not also possible this peak could be caused by a decomposition product from some other (e.g., 2,1-IN) precursor occurring during the analysis?

P18 L5: Unclear what is meant by signal/noise? It seems to me the natural unit here would be integrated signal (over Gaussian peak) per unit mol fraction for compound X divided by integrated signal (over Gaussian peak) per unit mol fraction?

P22 L3,4: 'Fig 4' should be 'Fig 3'?

Fig 6: These samples were collected trapped on Tenax? Perhaps include this information the caption.

SI:

P7 Caption L1: (3,4)-IN should be (4,3)-IN?

References:

Crounse, J. D., Teng, A., and Wennberg, P. O.: Experimental constraints on the distribution and fate of peroxy radicals formedin reactions of isoprene + OH + O2, presented at Atmospheric Chemical Mechanisms: Simple Models – Real World Complexities, University of California, Davis, USA, 10–12 December 2014.

Nguyen, T. B., et al: Overview of the Focused Isoprene eXperiment at the California Institute of Technology (FIXCIT): mechanistic chamber studies on the oxidation of

biogenic compounds, Atmos. Chem. Phys., 14, 13531-13549, doi:10.5194/acp-14-13531-2014, 2014.

Schwantes, et al., J. Phys. Chem. A, 2015, 119 (40), pp 10158–10171.

Teng, A. P., Crounse, J. D., Lee, L., St. Clair, J. M., Cohen, R. C., and Wennberg, P. O.: Hydroxy nitrate production in the OH-initiated oxidation of alkenes, Atmos. Chem. Phys., 15, 4297-4316, doi:10.5194/acp-15-4297-2015, 2015.

Xiong, F., et al : Observation of isoprene hydroxynitrates in the southeastern United States and implications for the fate of NOx, Atmos. Chem. Phys., 15, 11257-11272, doi:10.5194/acp-15-11257-2015, 2015.

---

## Author Comment (AC1) · 26 Jul 2016

**reviewer 1**

"Abstract: This section should contain more information that what is presented here in my opinion. It should provide the motivation for the study, a brief description of the method (or aim thereof), and a summary of the most relevant findings."

Changes : Abstract now reads:

According to atmospheric chemistry models isoprene nitrates play an important role in determining the ozone production efficiency of isoprene, however this is very poorly constrained through observations as isoprene nitrates have not been widely measured. Measurements have been severely restricted largely due to a limited ability to measure individual isoprene nitrate isomers. An instrument based on gas chromatography/mass spectrometry (GCMS) and the associated calibration methods are described for the speciated measurements of individual isoprene nitrate isomers. Five of the primary isoprene nitrates formed, in the presence of NOX, by reaction of isoprene with the hydroxyl radical (OH) in the Master Chemical Mechanism are identified using known isomers on two column phases, and are fully separated on the Rtx-200 column. Three primary isoprene nitrates from the reaction of isoprene with the nitrate radical (NO3) are identified after synthesis from the already identified analogous hydroxy nitrate. A Tenax adsorbent based trapping system allows the analysis of the majority of the known hydroxy and carbonyl primary isoprene nitrates, although not the (1,2)-IN isomer, under field-like levels of humidity, and showed no impact from typical ambient concentrations of NOX and ozone.

**P1L9 : Add "in the presence of NOx" to describe how isoprene nitrates are formed.**

Changes: has be added to the abstract in the revised manuscript

**Introduction: The words "recent" and "recently" are overused in the introduction. "Previous" is an alternative term that should be considered.**

Noted

Changes: We have removed/changed several uses of "recent" or "recently" in the introduction

P1L25: There are an extra set of parentheses around the references

Removed.

**P2L26:** Could you briefly elaborate on the "treatment of isoprene nitrates in the difference models"? Are they excluded, lumped, have different reactivities, etc.? Reply:

Most chemical schemes used in chemical transport models are heavily condensed mechanisms, often with species lumped according to different factors such as reactivity or structure, and contain far fewer species and reactions than the MCM. The isoprene degradation schemes used in transport models are similarly heavily condensed, having different numbers of species and reactions (e.g MOZART has 13 reactions, GOCART has 9 whilst the CRI has 22 reactions for the isoprene scheme). The isoprene nitrate sections of the model schemes are no different. The schemes vary with some not including isoprene nitrates at all, and of those that do include them, the rates of formation of isoprene nitrates vary along with how (or even if) photolysis, reaction with OH and deposition of isoprene nitrates are considered. Schemes that have lower yields of the nitrates tend to produce more ozone locally and less away from the source region (Squire et al, 2015). Similarly schemes that recycle the NOX from the nitrates effectively transports the NOX from source regions and the net effect is to produce more ozone than those schemes that do not (Emmerson and Evans, 2009).

**Changes :**

Introduction P1 L29 now reads: "The responses of different global chemistry transport models, in particular the production of ozone in the models, are sensitive to the isoprene reaction schemes (e.g. (Fiore et al. 2005; Wu et al. 2007)) and studies suggest that the yields of the of isoprene nitrates (INs) (Squire et al, 2015) and proportion of the NOX tied up in them that is recycled as opposed to lost (e.g. Emmerson and Evans, 2009) in the different models are the main factors in these discrepancies."

Included reference: Squire, O.J., A. T. Archibald, P. T. Griffiths, M. E. Jenkin, D. Smith, and J. A. Pyle, Atmos. Chem. Phys., 15, 5123–5143, , www.atmos-chem-phys.net/15/5123/2015/, doi:10.5194/acp-15-5123-2015, 2015.

**P2L25: It would help the reader if the list of species starting with acetone nitrate (NOA) were in the same order as that presented in Fig. 1 (i.e., (4,3)-IN, : : :, ending in NOA).**

Agree:

Changes: P2 L30. We have altered the order of the isomers in the text as suggested

**P2L27: List the remaining isomers to be more direct and reader friendly.**

We agree this would be better. Changes: We have fully listed the other isomers at P3 L1

**P3L15: Consider changing this section header to "Identification of Isoprene Nitrates via GC-MS"**

Changed as suggested.

**P3L16: Consider changing this section sub-header to "Chromatographic Methods"**

Changes: sub heading changed to "Chromatography"

P4L9: Figure S2.1 is really great. The authors should consider moving S2.1 into the main text in place of Fig. 2. In either figure (S2.1 and/or Fig. 2), it would be very helpful to put the molecular structures from Fig. 1 on each of the respective panels. It makes the mass spectra interpretation discussion (3.2.1 and 3.2.2) much easier to follow.

Excellent idea.

Changes: We have added the structures to the figure and we have replaced Figure 2 with Figure S2.1.

**P5L23: Consider changing this section sub-header to "Sample matrix and photochemistry experiments"**

Have changed as suggested.

P7L18: What is the volume of the drum used and the approximate concentrations of the analytes within the drum? How does the volume extracted for each sample (and sum of all samples) compare to the initial volume? Wall losses/effects would likely be more pronounced for lower pressure, lower concentration mixtures within a drum. Reply:

A rough estimate of the mixing ratios in the drum used for the precision and linearity test is included in section 5.3 of the paper, though we do not state the volume of the drum which was 100 litres. We did not determine the sensitivities of the GCMS system to most of the nitrates in the mixture, nor did we attempt any calibration during the period of the sampling tests, so we cannot report accurate species-specific mixing ratios. The individual sample volumes extracted from the drum represent 0.1 to 0.5% of the drum's volume with all samples from the precision and linearity tests totalling 4.2%. The sampled volume extracted from the drum was unavoidably replenished with lab air between each sample which would dilute the contents of the drum. The reviewer is correct to point out that wall effects will be important and it is the rapid loss of isoprene nitrates (presumably mainly losses on the walls) after initial introduction to the drum that prevents the production of predictable IN concentrations. Figure S3.3, shows that the observed mixing ratios inside the drum vary with temperature, suggesting that after these initial losses, there is a gas-phase/adsorbed-phase equilibrium established for the isoprene nitrates. This equilibrium will act to buffer the effects of dilution after sample extraction provided the volume extracted is only a few percent of that of the drum, which it is.

Changes:

P7 L23: Section 4 now includes the drum volume and approximate mixing ratios of the IN.

P11 L19: Section 5.3 now includes the volume of the drum.

John Crounse:

1) The authors state that there were problems for the isoprene hydroxynitrate trans-mission with longer columns (and warmer elution temperatures). Then it is stated the analyses were run on 30m instead of 60m columns, but it is not demonstrated that this completely eliminated the problem of hydroxynitrate decomposition or conversion. It is suggested here that the authors continue reducing their column length until results are reproducible. Additionally, I note that in our experience the 1-OH-2-ONO2 ISOPN isomer continues to suffer decomposition/isomerization problems in a metal-free low pressure GC-CIMS setup even on a 4-meter 0.53 megabore RTX-1701 column, with head-pressure at 150 mbar and tail pressure 40 mbar, under which it elutes at 60C. Additionally, and not without significant effort, we have not been able to trap the 1,2- externally and then transfer this quantitatively to the column without significant losses. Thus, our preferred method is to trap these compounds directly on the column phase at -15C. Care must be taken not to also condense water, as co-condensation of water and 1,2-ISOPN results in rapid hydrolysis of the 1,2-ISOPN.

Reply:

- a) We haven't proved that no decomposition occurs on the column for the INs which we tested, though the precision of the technique is good for all the tested nitrates (Z-(1,4)-IN and NOA were not present in the drum) which suggests that if decomposition is occurring within the system, it is constant and reproducible.
- b) We are now certain that (1,2)-IN isomer is lost on the system as described in the paper, since as the reviewer suggests the column length used dictates a high elution temperature as well as likely losses in the metal parts of the system and because of the high temperatures required for desorption from the Tenax trap. We have confirmed this by some recent tests on photolysed mixtures of an isoprene/NO/radical source using a 4 m column, cooled to 15 °C with a 2 ml sample directly injected onto the column using a PEEK valve and a PFA sample loop a process very similar to that used in PAN GC–ECD instruments. These injections clearly show a peak eluting before (4,3)-IN which is likely to be the (1,2)-IN isomer, although the other INs are not observed because of the poor detection limits expected with this method. Using any form of low-temperature pre-concentration or a column starting temperature of 5 °C or less results in the loss of both (1,2)-IN and (4,3)-IN, probably due to the condensation of water, something that is impossible to avoid in the real atmosphere.
- c) Regarding conversion in general, the analysis of pure samples of (4,3)-IN, E-(4,1)-IN, Z-(4,1)-IN and E-(1,4)-IN showed single isomers with no other isomers present to suggest in-system conversion. However, as reported in Bew et al (2016), Z-(1,4)-IN readily isomerises to the E-(1,4)-IN Isomer and all chromatograms of Z-(1,4)-IN showed a proportion of the E-(1,4)-IN isomer. Lee et al (2014) also report something similar despite using very different sampling and analysis conditions. Because the Z-(1,4)-IN isomer clearly isomerises outside of the analytical system, it is very difficult to disentangle the impact of analysis and trapping methods on the extent of isomerisation. Given

the readiness of Z-(1,4)-IN to isomerise, it is likely that some in-system reaction does occur and that we will thus overestimate the E-(1,4)/Z-(1,4) ratio to some extent. As suggested by the referee in a following comment, it is possible that Species X is a result of some conversion in the system, possibly from the (2,1)-IN, though without a pure sample of the (2,1)-IN with which to test this idea it remains speculation.

**Changes:**

P13 L7 modified to : "...the most likely explanation for our unexpected yields is that under the particular trapping and/or analytical conditions used here the (1,2)-IN isomer is lost before detection. The observation of what is believed to be the (1,2)-IN isomer using a direct injection method, cooled short column and no metal parts (see Supplementary info for details) supports this conclusion. "

We have included the following figure and text in the supplementary information for reference to support the belief that the Tenax based system results in the loss of the (1,2)-IN isomer, however the results are very preliminary and not comprehensive enough for inclusion in the main paper.

"Fig. S5.1 illustrates that using a direct injection method, short cooled column and no metal parts gives a peak eluting before (4,3)-IN which has similar ion fragments to the (4,3)-IN on our system. While we do not have a full mass spectrum of this (or identification from a pure sample), the elution prior to (4,3)-IN would suggest that this is (1,2)-IN based on the order of elution found by Nguyen et al. (2014). The fact that we do not see a peak with these ion fragments prior to (4,3)-IN when we use the Tenax based pre-concentration method as described in the main paper confirms that that (1,2)-IN is indeed lost on that system."

2) Order of elution for the hydroxynitrate isomers. In addition to the results published in Lee, et al, 2014, we now have NMR-supported identification of the 1,2-, c-4,1-, and t-4,1- ISOPN isomers. We find the elution order on RTX-1701 column to be 1,2-, 4,3-, c4,1-, (t4,1 and c1,4 together), and finally t1,4. These chromatograms from isoprene oxidation experiments are shown in Nguyen, et al., 2014, and the elution order for the five primary isoprene hydroxyl nitrates identified here is in agreement. In addition, Xiong, et al 2015, using I- CIMS shows a chromatogram from an OH+isoprene experiment, with a similar chromatographic technique. One primary problem that has remained through the years with the analysis of isoprene hydroxyl nitrates appears to be the very high instability of the 1,2- ISOPN. It appears to be thermally instable, decomposing and isomerizing to t1,4 isomer even at temperatures as low as 60C. Given the structural similarity of the 2,1-ISOPN to the 1,2-ISOPN and the general elution relationship for 1,2 vs 2,1 hydroxynitrates for a series of alkenes (Teng, et al, 2015). I would suggest that the 2,1-ISOPN isomer should elute even before the 1,2-ISOPN. Based on this, I suggest that the late-eluting species reported in this work arises from something other than (or a conversion product of) the 2,1-ISOPN species. Reply:

As discussed above, a direct injection onto a short, cooled column gives us a peak eluting before (4,3)-IN which Nguyen et al indicate is the (1,2)-IN isomer, and which has similar ion fragments to the (4,3)-IN on our system. While we do not have a full mass spectrum of this presumed (1,2)-IN isomer (or identification from a pure sample), its observation is in agreement with Nguyen and thus effectively rules out any possibility of species X being the (2,1)-IN isomer. Whether species X is a product of conversion from (2,1)-IN (or for that matter any other species) within the analytical system is impossible to say without the identity of species X being known or a pure sample of the (2,1)-IN isomer to test for conversion in the system.

**Changes:**

P12 L27 section 6 changed to-: "However, Nguyen et al. (2014) report that (2,1)-IN elutes before (4,3)-IN on the Rtx-1701 column, where as our species X elutes much later. The Nguyen study used a synthesised standard (although the synthesis and analytical data are as yet unpublished) to determine the retention order and thus precludes species X being the (2,1)-IN isomer. It is possible that species X is an isoprene-derived nitrate of some sort, possibly a conversion product of (2,1)-IN (or something else altogether), although without either the identity of species X being known or a pure sample of (2,1)-IN this remains speculation. "

3) The 1,2-ISOPN has a very fast hydrolysis rate. We've constrained (unpublished) its lifetime to be <5 seconds in D2O using NMR experiments. It is likely it also reacts rapidly in the condensed phase with other species, including itself. While, on a clean Teflon surface minimal H2O is adsorbed, it is possible that on a dirty Teflon surface, other low volatility hydrophilic material could be present, allowing for uptake of water, and thus allowing for a fast hydrolysis rate in a humid bag. I suggest that these experiments be tried with very dry air in a clean bag, using a GC method akin to those described in the references below (short column (2-4m), 8 sccm gas flow, sub-ambient pressure,

**such that 4,3-ISOPN elutes at <65C, no metal in sampling system or GCsystem (PEEK/PTFE/PFA seem okay), trap analytes directly on column at reduced temperatures (-15C)), and see if the same results are achieved.**

Reply:

It is almost certain that surface effects are significant in the small bags we used for the tests. However as discussed above when, as suggested by the reviewer, we used a very short column and removed all metal from the system (and using a direct injection rather than a Tenax trap) we were able to see a new peak elute just before the (4,3)-IN isomer which has fragment ions consistent with an IN and retention order consistent with (1,2)-IN as reported by Nguyen et al. We tested both dry and humid bags and saw similar results, but the tests were by no means comprehensive enough to draw any conclusions as to actual causes of variability in the system. The trapping method used by Nguyen et al will not be suitable for the analysis of real-world humid samples however, we will follow the advice of the reviewer and test the new direct injection methodology thoroughly under different conditions, but are unable to do so at present.

**P4 L5: Can the authors describe the energetics of the chemical (NI and PI) ionization**

**schemes?**

**Reply:**

The NI ionisation scheme uses 240 eV electrons and argon as the buffer gas to generate thermal electrons for electron capture. We have done (unpublished) tests comparing argon with methane as NI buffer gases on halogenated compounds and alkyl nitrates and have observed no difference in fragmentation patterns which suggests the energetics are very similar. At the usual source temperatures and buffer gas flows in an Agilent system, the thermal electrons will have an energy of approximately 0.06 eV. The PI analyses used methane as the reagent gas to produce  $CH_5^+$  as the major reagent ion, with H+ transfer as the most common ionisation reaction. Methane has a proton affinity of 552 kJ mol-1 (compare to water: 697 kJ mol-1) so proton transfer to methylnitrate (733 kJ mol-1) and nitrobenzene (800 kJ mol-1) will occur exothermically with energies of approximately 1.9eV and 2.6 eV respectively. We do not know the actual proton affinities of the INs, but ionisation energies of this magnitude seem reasonable.

**P5 L24-26: Suggest writing the isoprene and nitrite amounts as a mixing ratio, as this will be more useful to the reader (20ppmv, each).**

Changes: P5 L30: we have included approximations of the mixing ratios of the starting materials in the bags.

**P7 L19: Define MSD acronym.**

MSD has been replaced by "mass spectrometer" in the text and has been defined in the caption of Figure 4.

**P8 L29: Where did the NO2 come from? IN decomposition?**

Reply:

We presume the NO2 comes from IN decomposition since no NO or NO2 was introduced to the glass cubes or drum and the only known NO2 containing component was the isoprene nitrate. Measurements of NOY (hot quartz tube = NO2 + IN) in the cubes shortly after introduction of IN into the cube show that the concentrations of INs drop quickly after initial introduction. We did not measure the change in NO2 occurring during his time. The quantity of IN introduced to the calibration volume is poorly defined and any subsequent flushing or re-spiking of the cube to obtain the desired NOY signal means that each mixture has been treated differently and thus each dilution of an IN generated for calibrations contained different amounts of NO2.

Changes : P9 L6: modified the text to read "...different NO2 concentrations (presumably from decomposition of the IN in the drum)..."

Section 5.1: It seem to me that important inherent assumptions in this method of calibration include: a) decomposition of IN to NO2 in the cold quartz tube (or entire CL system) is small; b) reaction of IN with luminal is small. It would be useful to discuss the validity of these assumptions in more detail. It would be useful to include a figure of a CL experiment in the SI so the reader can get a feel for the time constants inherent with this method. What is the typical 'cold' CL signal compared to the IN signal?

**Reply:**

The reviewer is correct that there are implicit assumptions that the extent of decomposition of IN inside the CL system is small as is the direct reaction between IN and luminol. Figure S4.1 (below) shows the CL analysis of a dilution of (4,3)-IN with periods of the signal labelled A to F. In the paper we assume that the signal D-B represents  $NO_2$  in the calibration volume and that IN is determined by E-D, however any decomposition in the CL instrument itself and direct reaction with luminol would also contribute to D-B and thus give an underestimate of the true IN concentration.

In our experiments the magnitude of the D-B signal showed no strong correlation to either E-D or E-B (total  $NO_Y$ ) and varied between 6% and 30% of the total  $NO_Y$  signal. Direct reaction of the IN with luminol is possible, and PAN is well known to react quantitatively, although Hao et al (1994) report that gas-phase C6 alkyl- and hydroxy nitrates do not react with luminol directly. The residence times of gases, temperatures, solution compositions etc are constant in the CL system, so we would expect that any direct reaction would thus produce a signal that is a constant fraction of the IN. The lowest fraction of 6% for the D-B signal thus represents the maximum possible contribution from the direct reaction (if zero  $NO_2$  is present).

Similarly for decomposition within the CL system, if there is no direct reaction and  $NO_2$  is absent in the IN dilution, then internal decomposition can result in a maximum of 6% underestimation of the IN concentration. The wetted materials of the CL system are as inert as possible consisting of a short capillary column inlet, the quartz tube and PFA .Given the residence time of approximately 10 seconds inside the CL system (of which

Figure S4.1. Signal from the chemiluminescence analysis of a (4,3)-IN dilution in the glass calibration volume. A and F are  $NO_2$  calibrations (at 15.0 ppb), B is Zero air, C is lab air, D and E are the IN dilution sampled through cold (D) and hot quartz (E). E-D is the measured IN (22.7 ppb), D-B is the measured  $NO_2$  (2.7ppb).

Reference: Hao, C., Shepson, P.B., Drummond, J.W., Muthuramut, K., Gas Chromatographic Detector for Selective and Sensitive Detection of Atmospheric Organic Nitrates, Anal. Chem., 66, 3137-3143, 1994.

The discussion above and Figure S4.1 are now included in the supplementary information.

Section 5.2: Is it proper to interpret this procedure as contents of the cube were 'directly' sampled to the MS through a short piece of heated column? It would be naturalthen to compare the sensitivities determined in this manner to sensitivities determined in a similar manner with the Tenax trapping using the same CL reference, as this is needed to show that the transmission of the trapping method is high, and suitable for field samples.

Reply:

The calibration samples were performed with the same sample inlet, Tenax trap and in the same manner as the precision and linearity tests, only with smaller volumes than the precision tests because of the high concentrations in the dilutions. The heated sample inlet on the GCMS system is identical to that used on the CL system and the phrase "...to sample the contents directly" was intended to indicate that the end of the sample inlet was within the cube and that the sample had not been extracted through a union or other fitting which could cause losses.

Changes: P10 L27: We have stated explicitly that the calibrations were performed using the Tenax trap.

P11 L1: text now reads: "The cube's contents were sampled by the GCMS at 25 ml min-1 for 2-4 minutes through a heated capillary column inlet (0.53 mm ID Rtx-200 at 120  $^{\circ}$ C), 5 cm of which was inserted through the cube's inlet port to avoid sampling the cubes contents through an unheated fitting- the same sampling method used during the CL measurements. ".

**P11 L26, 31: It is stated here that the IN's discussed in this manuscript are stable on Tenax stored at -25C. This statement could use further support.**

Reply:

Pure (4,3)-IN, Z-(4,1), E-(4,1), E-(1,4)-al samples have been stored in our freezer at -25 °C for over a year without discolouration, and a year-old E-(1,4)-al-IN sample was used to obtain the NMR spectrum S6.1 in the supplementary information.

It is our experience that plastic materials (tubes/drums) that has been exposed to high concentrations of IN and subsequently left at room temperature show clear and consistent signs of the contaminant even after several weeks. This and the evidence that the IN persist in an aluminium drum at 30 °C for several months (and that there is a gas-phase/adsorbed-phase equilibrium in the drum) lead us to believe that the IN are somewhat stabilised when adsorbed and that low temperatures will enhance this stability. However we accept that this idea has not been demonstrated to be true.

changes:

P12 L6: text reads..." Many of the pure isoprene nitrates we synthesised are stable at -25  $^{\circ}$ C for at least several months which suggests it may be possible to store the tubes at -25  $^{\circ}$ C and use them in place of the main trap as an in-field multi-component standard. However this has yet to be confirmed experimentally."

**P12 L25-28: The 1,2-IN could also be lost through hydrolysis in the humid bag.**

Reply:

Possibly, but quick tests on a shorter column with a directly injected sample rather than Tenax trapping show what we think is (1,2)-IN as the largest IN peak, even in humid bags. However we have not been able to perform rigorous tests under different and controlled conditions to see what circumstances cause losses. What is clear is that we can only see the (1,2)-IN isomer when we don't use Tenax, metal valves, or long columns with high elution temperatures.

**P13 L13-16: Is it not also possible this peak could be caused by a decomposition product from some other (e.g., 2,1-**

**IN) precursor occurring during the analysis?**

Reply:

It is possible, but we have no way of testing or evaluating that hypothesis without either pure samples of the candidate isomers or having more information about species X.

**Changes:**

As discussed above, (P12 L27) now reads " However, Nguyen et al. (2014) report that (2,1)-IN elutes before (4,3)-IN on the Rtx-1701 column, where as our species X elutes much later. The Nguyen study used a synthesised standard (although the synthesis and analytical data are as yet unpublished) to determine the retention order and thus precludes species X being the (2,1)-IN isomer. It is possible that species X is an isoprene-derived nitrate of some sort, possibly a conversion product of (2,1)-IN (or something else altogether), although without either the identity of species X being known or a pure sample of (2,1)-IN this remains speculation. "

Section 7 now ends: (P13 L29) " The unidentified species X, produced during an attempt at synthesising (2,1)-IN, is also formed during the same photochemistry experiments in which the INs were observed. The limited evidence available suggests that it is a nitrated species and since it is only observed in photochemistry experiments when isoprene is present, it may be an isoprene-derived nitrate of some sort, although its identity and how it is formed (e.g. decomposition of (2,1)-IN) are entirely unknown."

**P18 L5: Unclear what is meant by signal/noise? It seems to me the natural unit here would be integrated signal (over Gaussian peak) per unit mol fraction for compound X divided by integrated signal (over Gaussian peak) per unit mol fraction?**

**Reply:**

The use of signal/noise (effectively peak height/baseline noise) rather than area was intended to account for two effects. The first is that when the GCMS is used in NI mode, the absolute signal for a given amount of substance decreases over time as the source becomes dirty and the filament degrades, so smaller peak heights (and thus also areas) are obtained with time. However, because the baseline noise is partly due to carrier contaminants, the absolute noise level also decreases with time which means S/N typically changes more slowly than peak height (or area) alone. During the period in which the data used in Table 2 were obtained, the mass spectrometer response had declined by about 10%. The other issue is that different masses have very different noise levels with m/z 46 having a noise level approximately 3x that of m/z 101 and twice that of m/z 99. Thus measuring the Z-(4,1)-IN isomer on m/z 99 will give a lower detection limit than measuring it on m/z46 despite its smaller peak area. However we accept that the units were not natural and lacked explanation.

Changes: We will present the table in the units suggested by the referee (see below for new table)

| Isoprene Nitrate | Sensitivity: Peak area/ppt of IN (ion) relative to n-butyl nitrate m/z 46 |         |        |        |
|------------------|---------------------------------------------------------------------------|---------|--------|--------|
|                  | m/z 46                                                                    | m/z 101 | m/z 99 | m/z 62 |
| (4,3)-IN         | 1.43                                                                      | 1.71    | 0.25   | 0.08   |
| Z-(4,1)-IN       | 1.89                                                                      | 0.48    | 1.06   | 0.43   |
| NOA              | 2.84                                                                      | -       | -      | 0.29   |

Table 2. Relative sensitivities of different INs to n-butyl nitrate, using different quantification ions. Relative sensitivity is calculated as peak area per unit mole fraction IN (ion) /peak area of n-butyl nitrate (m/z 46) per unit mole fraction n-butyl nitrate. It should be noted that the signal noise is typically a factor of 2 to 3 larger for m/z 46 than for the other ions.

**P22 L3,4: 'Fig 4' should be 'Fig 3'?**

Changed in manuscript

**Fig 6: These samples were collected trapped on Tenax? Perhaps include this information** the caption. Reply:**

The samples were indeed collected on Tenax in the same manner as described in section 4.

Changes: We have added this information to the caption.

**P7 Caption L1: (3,4)-IN should be (4,3)-IN?**

Yes, changed in manuscript